# POINT PROCESS FLOWS

## ABSTRACT

Event sequences can be modeled by temporal point processes (TPPs) to capture their asynchronous and probabilistic nature. We propose an intensity-free framework that directly models the point process as a non-parametric distribution by utilizing normalizing flows. This approach is capable of capturing highly complex temporal distributions and does not rely on restrictive parametric forms. Comparisons with state-of-the-art baseline models on both synthetic and challenging real-life datasets show that the proposed framework is effective at modeling the stochasticity of discrete event sequences.

## 1 INTRODUCTION

Data in real-life takes various forms. Event sequences, as a special form of data, are discrete events in continuous time. This type of data is prevalent in a broad spectrum of areas, for example, patient visits to hospitals, user behavior on social media, credit card transactions, etc. In this setting, each event is discrete, and the temporal dynamics of the events are complex and asynchronous. It is crucial to understand the characteristics and dynamics of such data, so that plausible future prediction, as well as other downstream applications, such as intervention or recommendation, can be performed. Despite recent success in modeling images, videos and texts with the power of deep neural networks (DNNs), the asynchronous and probabilistic nature of event sequence data makes it challenging to utilize the power of off-the-shelf DNN-based models.

Temporal point processes (TPPs; Daley & Vere-Jones 2007) provide us with an elegant and effective mathematical framework for modeling event sequences data. A temporal point process is defined as a stochastic process whose realizations consist of a list of events with their corresponding occurring times. These occurring times can either be real numbers from an index set (defined from prior knowledge) or sampled from an intensity function. While other time-series models learn temporal patterns synchronously (with each time-step being treated as an input to the model), TPP-based frameworks directly model the time intervals between events as random variables. With such a setup, it allows for modeling long sequences without vanishing gradients or costly memory issues.

Although the temporal point process has shown to be useful in modeling events sequences, it is usually not trivial to come up with a simple yet flexible intensity function. An intensity function encodes the rate an event occurs at a specified time-step. Poisson process (Kingman, 1992) has been a popular hand-crafted design for the intensity which assumes that events are independent of each other. More sophisticated design choices are investigated in the self-exciting (Hawkes, 1971) and self-correcting process (Isham & Westcott, 1979). The key contribution of these models is to find a functional form of intensity that fits data distribution well by making various parametric assumptions on the underlying generative process of the data. Although shown effective in modeling simple synthetic datasets, parametric assumptions make such frameworks lack the flexibility to model the generative process for real-life and complex data, hindering wider adoption of TPP-based frameworks.

Recently, learning the intensity function using recurrent neural networks (RNNs) to encode the history has received an increasing amount of attention (Du et al., 2016; Zhong et al., 2018; Mei & Eisner, 2017; Jing & Smola, 2017). In this line of work, history information is encoded and exploited in learning the intensity of the point process distribution. In this case, the explicit parametric assumption on the forms of the intensity functions is relaxed. However, the maximum likelihood training criteria on these models still requires the intensity function to be simple for the likelihood to be tractable. Recent work by Mehrasa et al. (2019) proposes a probabilistic framework based on variational autoencoders for modeling point process, further facilitating the stochastic generative

process. All the literature above is built upon explicit modeling of a temporal point process using the intensity function.

It is not necessary to explicitly model the intensity; a few works have tried to formulate TPP in an intensity-free manner. WGANTPP (Xiao et al., 2017; 2018) introduce an intensity-free framework for modeling the point process distribution using Wasserstein distance. The model is built upon a generative adversarial network (GAN). RLPP (Li et al., 2018) formulates this problem in a reinforcement learning framework and treats future event predictions as actions taken by an agent. Both of these models are optimized by trying to generate sequences of samples that are indistinguishable from the ground-truth sequences (by a discriminator in WGANTPP and policy learning in RLPP). Although these models are capable of generating realistic sequences, such training criteria fail to model the data distribution, resulting in intractable likelihood.

In this work, We propose a novel intensity-free point process model based on continuous normalizing flow and variational autoencoders. The proposed point process flow (PPF) model utilizes a recurrent variational autoencoder to encode the history of a given event sequence and makes probabilistic predictions on the next event. It preserves the non-parametric characteristics of point process distributions with normalizing flow. The predicted non-parametric point process distribution is capable of capturing complex time distributions of arbitrary shape, leading to more accurate modeling of event sequences. Extensive experiments are conducted on synthetic and real datasets to evaluate the performance of the proposed model. Experimental results show that our model is capable of capturing complex point process distributions as well as performing accurate stochastic forecast. The contributions are summarized as follows: (1) A novel intensity-free point process model built upon continuous normalizing flow. The proposed PPF is capable of capturing highly complex temporal distributions and does not rely on restrictive parametric forms; (2) PPF can be optimized by maximizing the exact likelihood using change of variable formula, relaxing the constraint in previous works that likelihood has to be tractable; (3) Evaluation on both synthetic and challenging real-life datasets shows improvement over the state-of-the-art point process models.

## 2 PRELIMINARIES

### 2.1 TEMPORAL POINT PROCESS

A temporal point process (TPP; Daley & Vere-Jones 2007) is a mathematical framework for modeling asynchronous sequences of actions. It is a stochastic process whose realization is a sequence of discrete events in time $t_{1:n} = \{t_1, t_2, ..., t_n\}$ where $t_n$ is the time when the $n^{\text{th}}$ event occurred.

A temporal point process distribution is modeled by specifying the probability density function of the time of the next event:

$$f(t|\mathcal{H}_t) = \lambda(t|\mathcal{H}_t) \exp\left\{ -\int_{t_{n-1}}^{t} \lambda(u|\mathcal{H}_u) \ \mathrm{d}u \right\}, \tag{1}$$

where the intensity function $\lambda(t|\mathcal{H}_t)$ is the conditional intensity function. It encodes the expected rate of event happening in a small area around $t$ and $\mathcal{H}_t = \{t_1, t_2, ..., t_{n-1}|t_{n-1} < t\}$ is the sequence of event times up to time $t$. Many works explored different design choices of intensity function to capture the phenomena of interest. Here we review two popular hand-crafted design choices:

**Poisson Process.** Poisson process (Kingman, 1992) is based on the assumption that events happen independent of each other where the intensity is a fixed positive constant $\lambda(t) = \lambda$ and $\lambda > 0$. In a more general case, $\lambda$ could be a function of time $\lambda(t|\mathcal{H}_t) = \lambda(t)$ but still independent of other events, which is called inhomogeneous Poisson process.

**Self-exciting Process (Hawkes process).** Self-exciting process (Hawkes, 1971) assumes that occurrence of an event increases the probability of other events happening in near future. Its intensity function has the functional form of $\lambda(t|\mathcal{H}_t) = \mu + \alpha \sum_{t_i < t} \exp(-(t - t_i))$, where $\mu$ and $\alpha$ are positive constants and $t_i < t$ are all the events happening before time t.

### 2.2 NORMALIZING FLOW

Normalizing flows are generative models that allow both density estimation and sampling. They map simple distributions to complex ones using bijective functions. Specifically, if our interest is

to estimate the density function $p_{\mathbf{X}}$ of a random vector $\mathbf{X} \in \mathbb{R}^d$, then normalizing flows assume $\mathbf{X} = g_\theta(\mathbf{Z})$, where $g_\theta : \mathbb{R}^d \to \mathbb{R}^d$ is a bijective function, and $\mathbf{Z} \in \mathbb{R}^d$ is a random vector with a tractable density function $p_{\mathbf{Z}}$. We further denote the inverse of $g_\theta$ by $f_\theta$. On one hand, the probability density function can be evaluated using the change of variables formula:

$$p_{\mathbf{X}}(\mathbf{x}) = p_{\mathbf{Z}}(f_\theta(\mathbf{x})) \left| \det \left( \frac{\partial f_\theta}{\partial \mathbf{x}} \right) \right|, \tag{2}$$

where $\partial f_\theta / \partial \mathbf{x}$ denotes the Jacobian matrix of $f_\theta$. On the other hand, sampling from $p_{\mathbf{X}}$ can be done by first drawing a sample from the simple distribution $\mathbf{z} \sim p_{\mathbf{Z}}$, and then apply the bijection $\mathbf{x} = g_\theta(\mathbf{z})$.

Given the expressive power of deep neural networks, it is natural to construct $g_\theta$ as a neural network. However, it requires the bijection $g_\theta$ to be invertible, and the determinant of the Jacobian matrix should be efficient to compute. Several methods have been proposed along this research direction (Rezende & Mohamed, 2015; Dinh et al., 2014; 2017; Kingma et al., 2016; Kingma & Dhariwal, 2018; Papamakarios et al., 2017). An extensive overview of normalizing flow models is given by Kobyzev et al. (2019).

## 2.3 CONTINUOUS NORMALIZING FLOW

From a dynamical systems perspective, the residual network can be regarded as the discretization of an ordinary differential equation (ODE; Haber & Ruthotto 2017; Chang et al. 2018; Lu et al. 2018). Inspired by that, Chen et al. (2018) propose neural ODE, where the continuous dynamics of hidden units is parameterized using an ordinary differential equation specified by a neural network:

$$\frac{d\boldsymbol{z}(t)}{dt} = h(\boldsymbol{z}(t), t, \theta). \tag{3}$$

The neural ODE can be used to construct a continuous normalizing flow. The invertibility is naturally guaranteed by the theorem of the existence and uniqueness of the solution of the ODE. Furthermore, using the instantaneous change of variables formula, similar to Equation 2, the log-density can be evaluated by solving the following ODE:

$$\frac{\partial \log p(z(t))}{\partial t} = -\mathrm{Tr}\left( \frac{\partial h}{\partial z(t)} \right). \tag{4}$$

Grathwohl et al. (2019) propose an improved version of neural ODE, named FFJORD, which has lower computational cost by using an unbiased stochastic estimation of the trace of a matrix.

## 3 PROPOSED FRAMEWORK

We propose an intensity-free flow framework to model the timing of events in point process sequences. More specifically, we learn a non-parametric distribution over the timing of asynchronous event sequences by transforming a simple base probability density through continuous normalizing flow, *i.e.*, a series of invertible transformations. With our proposed framework, we are able to model complex point process distributions without making any assumption on the functional form of the distribution while being able to evaluate the likelihood of sequences under our model.

### 3.1 NON-PARAMETRIC POINT PROCESS FLOWS

Let the input be a sequence of asynchronous events $\{t_1, t_2, ...\}$, where $t_i \in \mathbb{R}_+$ represents the starting time of the $i$-th event. We define the inter-arrival time $\tau_n$ as the time difference between the starting time of events $t_{n-1}$ and $t_n$. Our goal is to model the distribution over inter-arrival time $\tau_n$ given the past history of events inter-arrival times $\tau_{1:n-1}$, *i.e.*, learning to model the conditional distribution $p(\tau_n | \tau_{1:n-1})$.

Our approach is to construct the distribution over inter-arrival time $\tau_n$ by transforming a simple base distribution through normalizing flow transformations. At time-step $n$, we assume that inter-arrival time $\tau_n$ was generated by first sampling from a simple distribution $p(z_n)$ and then transforming the drawn sample $z_n$ through an invertible transformation $g_\theta : \mathbb{R} \to \mathbb{R}_+$ parametrized by $\theta$ :

$$z_n \sim p(z_n), \quad \tau_n = g_\theta(z_n). \tag{5}$$

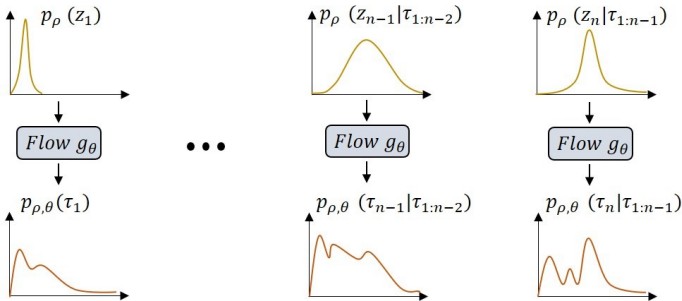

Figure 1: This figure shows the overall structure of our non-parametric point process modeling via normalizing flow. We learn a non-parametric distribution over the inter-arrival time of asynchronous event sequences by transforming a base probability density which is conditioned on the past history through normalizing flow transformations.

With this assumption and the change of variable formula discussed in subsection 2.2, we can write the distribution over inter-arrival time $\tau_n$ as:

$$p_\theta(\tau_n) = p(z_n) \left| \frac{\partial f_\theta}{\partial \tau_n} \right|, \tag{6}$$

where $f_\theta(\tau_n) = g_\theta^{-1}(\tau_n) = z_n$, and the scalar Jacobin value $df_\theta/d\tau_n$ shows the changes in the density when moving from $\tau_n$ to $z_n$. We dropped the determinant term in the change of variable formula, because in our case, the inter-arrival time $\tau_n$ is a one-dimensional variable. With this formulation, we are able to model the inter-arrival time distribution, without any specific assumption on the functional form of the distribution. Exact samples of the inter-arrival time distribution can be obtained by sampling from the base distribution $z_n \sim p(z_n)$ and transforming it through flow transformation $\tau_n = g_\theta(z_n)$. We are also able to compute the exact likelihood of $\tau_n$ by computing the likelihood of $f_\theta(\tau_n)$ and multiplying it with the associated Jacobian term $\left| \frac{\partial f_\theta}{\partial \tau_n} \right|$.

Current formulation models the inter-arrival distribution of each time-step independent of past history. The future event timing might depend on previous events in a very complex way, so its important to take the history information into consideration while modeling future events. To capture this dependency, we adapt our formulation to construct the point process distribution by learning normalizing flow parameters conditioned on the history. More specifically, we learn the parameters of flow base distribution by a time-dependent model parametrized by $\rho$ that encodes history and provides the conditional base distribution $p_\rho(z_n|\tau_{1:n-1})$ at each time-step. The overal procedure can be seen in Figure 1. In our framework, the base distribution is assumed to follow a Gaussian distribution:

$$p_\rho(z_n|\tau_{1:n-1}) = \mathcal{N}(\mu_{\rho_n}, \sigma_{\rho_n}^2), \tag{7}$$

where $(\mu_{\rho_n}, \sigma_{\rho_n}^2)$ are the parameters of the base distribution. These parameters can be obtained by encoding the history $\tau_{1:n-1}$ using various approaches. In this work, we propose two approaches to construct the base distribution of the flow: 1) Base distribution with deterministic parameters. 2) Base distribution with stochastic parameters. In the following sections, we describe each approach in more details.

## 3.2 BASE DISTRIBUTION WITH DETERMINISTIC PARAMETERS

As discussed in subsection 3.1, we aim to model the conditional distribution $p(\tau_n|\tau_{1:n-1})$ using history information in learning the base distribution parameters. Recurrent neural networks (RNNs) have shown to be powerful deterministic models in capturing temporal dependencies. Recent works adapt RNNs as a non-linear mapping of the history to the intensity function to define temporal point process distributions (Du et al., 2016; Zhong et al., 2018; Mei & Eisner, 2017). As a first attempt, we use RNNs to learn the parameters of the base distribution of the flow using the history information.

Figure 2 part (a) illustrates the overall structure of our model. To construct the conditional distribution $p_{\theta,\rho}(\tau_n|\tau_{1:n-1})$, the RNN takes the history of the past inter-arrival times $\tau_{1:n-1}$ and produces

(a) Deterministic Approach

Flow Module

$\tau_{1:n-1} \rightarrow$ [ $RNN_\rho$ ] $\rightarrow p_\rho(z_n^\tau|\tau_{1:n-1}) \rightarrow$ [ $g_\theta$ ] $\rightarrow p_{\theta,\rho}(\tau_n|\tau_{1:n-1})$

(b) Probabilistic Approach

Flow Module

$\tau_{1:n-1} \rightarrow$ [ $Enc_\psi$ ] $\rightarrow \mathcal{N}(\mu_{\psi_n}, \sigma_{\psi_n}) \sim z_n^{vae} \rightarrow$ [ $Dec_\rho$ ] $\rightarrow p_\rho(z_n^\tau|z_n^{vae}) \rightarrow$ [ $g_\theta$ ] $\longrightarrow p_{\theta,\rho}(\tau_n|z_n^{vae})$

Figure 2: Part (a) shows the deterministic approach of utilizing RNNs for predicting conditional distribution $p_{\theta,\rho}(\tau_i|\tau_{1:i-1})$. RNN encodes history into the base distribution, then it gets transformed to the target distribution by flow transformation $g_\theta$. Part (b) shows the generation phase of incorporating the flow module in a probabilistic framework. During generation, the prior network gets the history and output the latent space distribution for the next time-step. Then a sample of this distribution is passed to the decoder which generates the non-parametric distribution over the inter-arrival time of next time-step by incorporating the flow module.

the conditional base distribution $p_\rho(z_n|\tau_{1:n-1}) = \mathcal{N}(\mu_{\rho_n}, \sigma_{\rho_n}^2)$ for the next time-step. Then, the base distribution is transformed into the conditional inter-arrival time distribution over $\tau_n$ through normalizing flow transformations $g_\theta$. The RNN is jointly optimized with the flow module by maximizing the log-likelihood of observed sequence $\tau_{1:N}$ under the predicted distribution:

$$\mathcal{L}_{\theta,\rho}(\tau_{1:N}) = \sum_{i=1}^{N} \log p_{\theta,\rho}(\tau_i|\tau_{1:i-1}) = \sum_{i=0}^{N} \log p_\rho(z_i|\tau_{1:i-1}) + \log\left|\frac{\partial f_\theta}{\partial \tau_i}\right|. \tag{8}$$

### 3.3 BASE DISTRIBUTION WITH PROBABILISTIC PARAMETERS

It is known that there is a trade-off between the complexity of the bijective transformation and the form of base distribution (Jaini et al., 2019). With the complexity of the bijective transformation fixed, a more flexible base distribution will lead to a more expressive model. In our proposed framework, the fact that flow transformations are shared across time-steps and the true underlying distribution across time-steps might vary a lot, makes our model more sensitive to the choice of base distribution family. We believe that, if we choose to model base distributions as Gaussian distributions with deterministic parameters, the bijective transformation might not be able to estimate underlying distributions well. We further support our claim by proving Proposition 1 which, intuitively speaking, says more flexible base-distribution yields more expressive model.

Motivated by this, our second move is to have a more flexible base-distribution where the parameters are probabilistic. In order to achieve this, we utilize the variational auto-encoder (VAE; Kingma & Welling 2014) paradigm in modeling the conditional base-distributions. To better illustrate the importance of having more flexible base-distribution, we provide a motivating example in Appendix A.

To avoid confusion, at time-step $n$, we use the notation $z_n^\tau$ for the random variable of the normalizing flow base distribution and $z_n^{vae}$ to refer to the VAE latent space. We start by explaining the generation phase, *i.e.*, how the distributions over inter-arrival time $\tau_n$ are generated by stacking the normalizing flow module on top of the VAE backbone and then describing the training process.

**Generation.** Figure 2 part (b) shows an overview of the generation process. Here, we adapt a recurrent VAE framework consisting of a time-variant prior network parametrized by $\psi$ which takes the history of past actions $\tau_{1:n-1}$ and provides the latent distribution $p_\psi(z_n^{vae}|\tau_{1:n-1})$. Then, a sample of this distribution is passed to the VAE's decoder which produces a non-parametric distribution over the inter-arrival time $\tau_n$ by first generating the normalizing flow base distribution $p_\rho(z_n^\tau|z_n^{vae})$ and then transforming it through flow transformation $g_\theta$. By applying the change of variable formula discussed in Equation 6, we can write the distribution over inter-arrival time $\tau_n$ as:

$$p_{\theta,\rho}(\tau_n|z_n^{vae}) = p_\rho(z_n^\tau|z_n^{vae})\left|\frac{\partial f_\theta}{\partial \tau_n}\right|. \tag{9}$$

**Training.** At time-step $n$ of training, the VAE module takes the sequence of inter-arrival times $\tau_{1:n}$ to approximate the true distribution over the latent space $z_n^{vae}$ via the help of the recurrent inference network $q_\phi(z_n^{vae}|\tau_{1:n})$ which is parametrized with $\phi$. A time-dependent prior network is also adapted to help the model to take use of history information in generation phase $p_\psi(z_n^{vae}|\tau_{1:n-1})$. Both prior and posterior distributions are assumed to follow conditional multivariate Gaussian distributions with diagonal covariance:

$$p_\psi(z_n^{vae}|\tau_{1:n-1}) = \mathcal{N}(\mu_{\psi_n}, \Sigma_{\psi_n}), \tag{10}$$

$$q_\phi(z_n^{vae}|\tau_{1:n}) = \mathcal{N}(\mu_{\phi_n}, \Sigma_{\phi_n}). \tag{11}$$

At each time-step during training, a latent code $z_n^{vae}$ is taken from the posterior and is passed to the decoder which aims to generate the conditional distribution $p_{\theta,\rho}(\tau_n|z_n^{vae})$. The VAE backbone is jointly trained with the flow module by optimizing the variational lower bound using the re-parameterization trick (Kingma & Welling, 2014):

$$\mathcal{L}_{\theta,\phi,\psi,\rho}(\tau_{1:N}) = \sum_{n=1}^{N}(\mathbb{E}_{q_\phi(z_n^{vae}|\tau_{1:n})}[\log p_{\theta,\rho}(\tau_n|z_n^{vae})] \tag{12}$$
$$- D_{KL}(q_\phi(z_n^{vae}|\tau_{1:n})||p_\psi(z_n^{vae}|\tau_{1:n-1}))),$$

where we compute the log-likelihood term $\log p_{\theta,\rho}(\tau_n|z_n^{vae})$ by applying the change of variable formula of Equation 9.

## 4 EVALUATION

To show the effectiveness of our non-parametric approach, we evaluate the performance of our model on synthetic and real-world datasets and compare it with the state-of-the-art point process models. Please refer to Appendix A for architecture and implementation details.

### 4.1 DATASETS AND BASELINES

**Synthetic Datasets.** We create three types of synthetic datasets as follow: **(I) Inhomogeneous Poisson Process (IP)** defines the intensity as a function of time but independent of the history. We simulate sequences of IP process with $\lambda(t) = \sum_{i=1}^{k} \alpha_i (2\pi\sigma_i^2)^{-1/2} \exp(-(t-c_i)^2/\sigma_i^2)$ where $k = 6$, $\alpha = (14, 18, 13, 17, 10, 13)$, $c = (3, 6, 9, 12, 15, 18)$ and $\sigma = (5, 5, 5, 5, 5, 5)$. **(II) Self-exciting Process (SE)** assumes that the occurrence of an event increases the probability of other events happening in the near future. It is characterized by $\lambda(t) = \mu + \beta\sum_{t_i<t} g(t - t_i)$, where in our case $g(t) = \exp(-t)$, $\mu = 1.0$, and $\beta = 0.8$. **(III) IP + SE** is created by combining the simulated data from the self-exciting process and the inhomogeneous process. For each of IP and SE, we generate 20000 sequences, where the length of each is 60, and split the sequences into train, validation and test sets with proportions of 0.7, 0.1, 0.2, respectively.

**Real-world Datasets.** We also evaluate our models on real datasets that cover the areas of social media, healthcare, and human activity as follow: **(I) LinkedIn** data is collected from over 3000 LinkedIn accounts and contains their job-hopping records with information including the time and company. Our model predicts the time-interval before a user's next job-hopping. After pruning users with only one job-hopping record, we collect 2439 sequences. **(II) MIMIC** (Medical Information Mart for Intensive Care III; Johnson et al. 2016; Pollard 2016) is a publicly available, large-scale dataset which contains the medical records of more than 40000 anonymous patients. Our method models the inter-arrival time of patients' admissions to hospital. We keep the record of patients who have at least three visits to hospitals and collected 2377 sequences. **(III) Breakfast** dataset (Kuehne et al., 2014) contains 1712 videos with 48 action classes related to breakfast preparation. On this dataset, we model the inter-arrival times of the actions as well as actions categories. We explain this extension in more detail in subsection 4.2. For the **LinkedIn** and **MIMIC** dataset, we also split the dataset into train, validation and test sets with proportions of 0.7, 0.1, 0.2, respectively. For the **Breakfast** dataset, we use the standard train and test split proposed by Kuehne et al. (2014).

**Baselines.** We compare our proposed flow-based approach with the state-of-the-art point process models: **(I) APP-LSTM**[1] is an LSTM that takes the history of past events and predicts the inter-arrival time distribution for the next time-step by mapping the history into the intensity of a point

---

[1]This baseline has comparable performance to Mei & Eisner (2017); Du et al. (2016).

| Dataset | Model | | | |
|---------|-------|------|------|------|
|         | APP-LSTM | PPF-D | APP-VAE | PPF-P |
| IP      | $-2.942$ | $-1.857$ | $\geqslant 0.408$ | $\geqslant \mathbf{0.499}$ |
| SE      | $-2.990$ | $-1.615$ | $\geqslant 0.562$ | $\geqslant \mathbf{0.636}$ |
| IP+SE   | $-2.978$ | $-1.507$ | $\geqslant 0.476$ | $\geqslant \mathbf{0.566}$ |
| LinkedIn | $-0.795$ | $0.973$ | $\geqslant -1.713$ | $\geqslant \mathbf{2.678}$ |
| MIMIC   | $-1.962$ | $-0.498$ | $\geqslant -1.200$ | $\geqslant \mathbf{1.696}$ |

Table 1: **Log-likelihood Comparison.** LL is reported for synthetic and real datasets.

process distribution. We train it by maximizing the likelihood of observed sequences under the predicted distribution. This deterministic baseline has comparable performance to Du et al. (2016). It only differs in the way that intensity is defined; unlike Du et al. (2016), its intensity doesn't explicitly depend on time. Zhong et al. (2018) compare these two design choices, and implicit dependence was shown to be more effective in modeling point process distribution.

**(II) APP-VAE** (Mehrasa et al., 2019) is a latent variable framework for modeling marked temporal point process. The model makes conditional predictions by learning a conditional latent space. Given a history of past actions, APP-VAE generates two distributions for the next action: one over its timing (by predicting the conditional intensity and using it to define point process distributions) and one over its category. On breakfast dataset, we use their original setup with the use of both time and mark data to have a fair comparison with APP-VAE; using mark data could help better capturing the temporal dependencies. For the rest of the datasets, we modify their approach to predict the time distribution only.[2]

### 4.2 RESULTS

**Log-likelihood Comparison.** We report log-likelihood (LL) of test sequences across all models. For our PPF model with the probabilistic approach in learning base distribution parameters (PPF-P; introduced in subsection 3.3) and APP-VAE baseline, we report the importance weighted autoencoder (IWAE) bound, which is a lower bound of the real log-likelihood. To compute IWAE, at each time-step, we draw 1500 samples from the VAE's posterior distribution and follow the standard procedure for computing IWAE. We report the average of log-likelihood along all the time-steps of all sequences in the test dataset. The experimental results are shown in Table 1. The results indicate the better capability of our normalizing flow-based approaches at modeling point process sequence data, especially the real-world data with complicated underlying distributions. Our probabilistic PPF-P approach robustly outperforms state-of-the-art intensity-based baselines across all the datasets. Without any assumption on the functional form of intensity, we are able to model the point process distribution effectively. Our probabilistic approach also has a better performance in comparison to our deterministic approach introduced in subsection 3.2 (PPF-D). This demonstrates the advantages of using a more flexible base distribution in the flow in the probabilistic approach.

**Point Estimate Comparison.** We also report the mean absolute error (MAE) to evaluate the performance of our model in estimating future events timing. The MAE between the samples of predicted time distribution and the ground-truth is reported. To compute MAE for PPF-P and APP-VAE, at time-step $i$, we have two stages of sampling: (1) First, we draw samples from the prior distribution $z_i^{vae} \sim p_\psi(z_i^{vae}|\tau_{1:i-1})$. Then, we pass each to the decoder and (2) draw samples from each predicted distribution $\tau_i \sim p_\rho(\tau_i|z_i^{vae})$. The MAE computation at time-step $i$ is as follow:

$$\mathbb{E}_{z_i^{vae} \sim p_\psi(z_i^{vae}|\tau_{1:i-1})}\left[\mathbb{E}_{\tau_i \sim p(\tau_i|z_i^{vae})}\left(|\tau_i - \tau_i^*|\right)\right], \tag{13}$$

where $\tau_i^*$ is the ground-truth inter-arrival at time-step $i$. We follow a similar procedure for computing the MAE for the deterministic approaches:

$$\mathbb{E}_{\tau_i \sim p(\tau_i|\tau_{1:i-1})}\left(|\tau_i - \tau_i^*|\right). \tag{14}$$

For PPF-P and APP-VAE, to estimate the expected error, we draw 100 samples from prior distribution $p_\psi(z_i^{vae}|\tau_{1:i-1})$ and 15 samples from each predicted base distribution $p_\theta(\tau_i|z_i^{vae})$. For

---

[2]We drop the use of mark data as input and accordingly omit the likelihood calculation of action category distribution from the optimization term.

| Dataset | Model | | | |
|---|---|---|---|---|
| | APP-LSTM | PPF-D | APP-VAE | PPF-P |
| IP | 6.765 | 4.7759 | 0.279 | **0.278** |
| SE | 7.360 | 4.4205 | **0.290** | 0.297 |
| IP+SE | 7.163 | 4.0525 | **0.288** | 0.299 |
| LinkedIn | 2.522 | 2.048 | 2.495 | **1.799** |
| MIMIC | 23.531 | **17.709** | 27.479 | 26.047 |

Table 2: **Mean Absolute Error Comparison.** MAE is reported for synthetic and real datasets.

| Model | LL | MAE | Accuracy |
|---|---|---|---|
| APP-LSTM | $-8.099$ | 239.624 | 59.594 |
| PPF-D | $-7.637$ | 251.337 | 61.174 |
| APP-VAE | $\geqslant -6.463$ | 244.019 | 62.190 |
| PPF-P | $\geqslant -\mathbf{6.342}$ | **204.913** | **62.528** |

Table 3: Comparison of log-likelihood, MAE, and accuracy on Breakfast dataset.

PPF-D and APP-LSTM, we sample 1500 predictions from the output distributions $p(\tau_i|\tau_{1:i-1})$ at each step. For our PPF approaches (both deterministic and probabilistic), the corresponding samples of predicted inter-arrival time distribution are obtained using Equation 5. We report the average of MAE along all the time-steps of all sequences in the test dataset. Table 2 shows the experimental results for MAE metric. We can see that our PPF-P approach is comparable to the APP-VAE baseline on the synthetic datasets. On the more challenging real datasets, our PPF-based frameworks consistenlty outperforms baseline models (PPF-D on MIMIC, and PPF-P on LinkedIn). The better log-likelihood estimations is also conformed by lower/competitive MAE which reflects the better quality of generated samples from our PPF approaches.

**Extension to the Marked Temporal Point Process.** On Breakfast dataset, for our model to learn a more powerful encoding of history information, we extend our approach to marked point process which models both the inter-arrival time distribution of future event and also the distribution over its category. At time-step $i$, given the history of past events including both time and mark information, in addition to modeling the time distribution of event at time-step $i + 1$, we also model its category distribution. Here, we assume that action category follows a multinomial distribution and accordingly, the log-likelihood of action category distribution modeling is added to training objective and evaluation criterion of our deterministic and probabilistic PPF models.[3] APP-LSTM is also extended to the marked case similar to Du et al. (2016). For the experiments on Breakfast dataset, in addition to MAE, we also report the accuracy of predicting the next action category. To compute accuracy at time-step $i$, for probabilistic approaches, we draw 100 samples from the prior distribution $p_\psi(z_i^{vae}|\tau_{1:i-1})$ and for each predicted category distribution, we select the action category with maximum probability as the predicted class. For each time-step, the most frequently predicted type is reported as the model's prediction. Table 3 shows the experimental results of comparing log-likelihood, MAE, and accuracy on Breakfast dataset. This dataset is much more challenging in comparison to LinkedIn and Mimic datasets, because of it contains various types of actions. We can see that our probabilistic approach has a better performance in all the metrics which shows the effectiveness of our proposed model in capturing the underlying point process distribution.

## 5 CONCLUSION

In this paper, we propose Point Process Flows (PPF), an intensity-free framework that directly models the point process as a non-parametric distribution by utilizing normalizing flows. The proposed model is capable of capturing arbitrary complex time distributions as well as performing stochastic future prediction. The proposed PPF can be optimized by maximizing the likelihood using change of variable formula, relaxing the strict tractable likelihood constraint in previous works. Extensive

---

[3]For our deterministic/probabilistic approach, we assume that at each time-step given the history/latent-code, time and category are independent.

evaluation on both synthetic and challenging real-like datasets shows significant improvement over baseline models.

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

## A APPENDIX

### A.1 ARCHITECTURE

The overall architectures are illustrated in Figure 2. For the deterministic approach introduced in subsection 3.2, a long short-term memory (LSTM; Hochreiter & Schmidhuber 1997) network is used to model the conditional distribution $p_\rho(z_n | \tau_{1:n-1}) = \mathcal{N}(\mu_{\rho_n}, \sigma^2_{\rho_n})$. The flow module $g_\theta$ is a neural ODE model described in subsection 2.3, where the derivative function $h(\cdot)$ is modeled by a multilayer perceptron (MLP).

For the probabilistic approach in subsection 3.3, both the prior distribution $p_\psi(z_n^{vae} | \tau_{1:n-1})$ and the approximate posterior distribution $q_\phi(z_n^{vae} | \tau_{1:n})$ are modeled by LSTMs. The log-likelihood term $\log p_{\theta,\rho}(\tau_n | z_n^{vae})$ is computed in two steps. First, a decoder network maps the latent variable $z_n^{vae}$ to a base distribution $p_\rho(z_n^\tau | z_n^{vae})$, which is also a normal distribution. The decoder network is a MLP that outputs the parameters of the base distribution. After that, the flow module $g_\theta$ generates the distribution of the inter-arrival time $p_{\theta,\rho}(\tau_n | z_n^{vae})$. The architecture of $g_\theta$ is the same as in the deterministic approach.

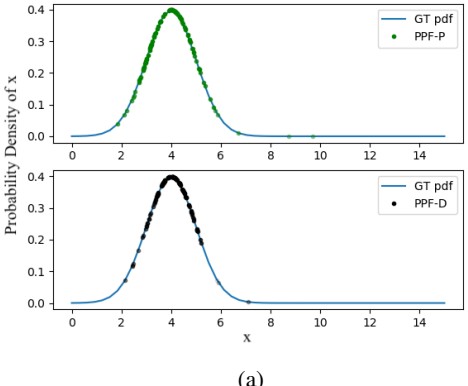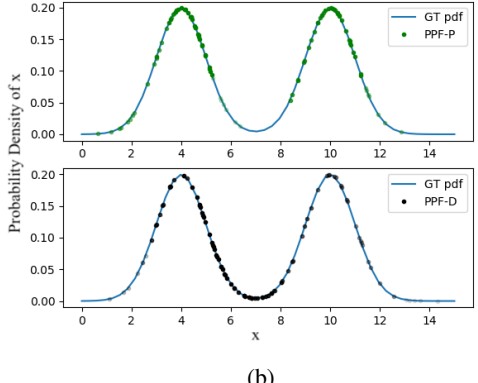

|     |     |
| --- | --- |
| (a) | (b) |

Figure 3: Part (a) shows the results of modeling conditional distribution $P(x_i|x_{1:i-1}) = \mathcal{N}(4, 1)$. Part (b) shows the results modeling the conditional distribution $P(x_{i+1}|x_{1:i}) = .5 * \mathcal{N}(4, 1) + .5 * \mathcal{N}(10, 1)$. The top/bottom figure of each sub-figure shows the generated samples by PPF-P/PPF-D. As we can see, PPF-D is not able to handle this case very well becausethere could be no one-to-one transformation that can map two Gaussian base distributions exactly into a uni-modal and a multi-modal distribution respectively at the same time. However, PPF-P perform much better at modelling the true underlying distributions.

## A.2 IMPLEMENTATION DETAILS

For the LSTM cells, we choose hidden size to be 128 for the synthetic data and **Breakfast** dataset, 64 for **LinkedIn** and **MIMIC** datasets. The dimension of the latent space of VAE models is set to be 256 for **Breakfast** and synthetic datasets and 64 for **LinkedIn** and **MIMIC**. For PPF-P model, the latent code was decoded into the mean and variance of the base distribution by two separate decoders, each with two hidden layers of size 256. For all continuous normalizing modules, we use one block of network with 3 hidden layers of 64 dimension. We use Adam optimizer (Kingma & Ba, 2015) for all models with a learning rate of 0.001.

## A.3 A MOTIVATING EXAMPLE FOR PPF-P

When using flow techniques for density estimation, the expressiveness of the model is not only limited by the complexity of normalizing flow transformations, but also by the class of base-distributions. In our proposed framework, the fact that the flow transformations are shared across time-steps and the underlying distribution across time-steps might vary a lot makes our model more sensitive to the choice of the base-distribution family. By introducing a latent variable $z_n^{vae}$ such that the $z_n^{\tau}$ follows different Gaussian distributions conditioned on different samples of $z_n^{vae}$, the distributions of $z_n^{\tau}$ after marginalizing $z_n^{vae}$ becomes highly flexible.

To motivate this argument, we make the following proposition, show its proof and substantiate it with experiment results on PPF-P and PPF-D models.

**Proposition 1.** *Let $f : \mathbb{R} \to \mathbb{R}$ be a bijective singular mapping that satisfies the following: $\mathbf{z} \sim \mathcal{N}(\mu_0, \sigma_0^2)$ and $\mathbf{x} = f(\mathbf{z})$ follows a mixture of Gaussian distribution of two components, $\mathcal{N}(\mu_1, \sigma_1^2)$ and $\mathcal{N}(\mu_2, \sigma_2^2)$, with weights $a$ and $1 - a$ for some $a \in (0, 1)$. There exists $i \in \{1, 2\}$, such that if $\mathbf{x}$ is sampled from component $i$ of the Gaussian mixture distribution $\mathbf{x} \sim \mathcal{N}(\mu_i, \sigma_i^2)$, $f^{-1}(\mathbf{x})$ does not follow a Gaussian distribution.*

In summary, Proposition 1 says if a normalizing flow can maps a Gaussian distribution to a mixture of Gaussian distributions, which is multi-modal, we can not obtain one of the mixture distribution's components by applying the same normalizing flow to another Gaussian distribution. Continuous normalizing flow defines a continuous bijective mapping from $\mathbb{R}$ to $\mathbb{R}$ and therefore it must be singular increasing or decreasing. The normalizing flow we used is shared for all time steps and its derivative is independent of the parameters of the base distribution. It is also worth noting that this

proposition can be extended to scenarios of any Gaussian mixture distributions with finite components.

*Proof.* Without loss of generality, consider the following two cases:

**Case 1** $\sigma_1^2 > \sigma_2^2$. We show that the inverse mapping of $\mathbf{x} \sim \mathcal{N}(\mu_1, \sigma_1^2)$ is not a Gaussian by contradiction. Suppose $\mathbf{z} = f^{-1}(\mathbf{x})$ follows a Gaussian distribution $\mathcal{N}(\mu_3, \sigma_3^2)$. By our assumptions and the change of varaible theorems we have the following

$$\frac{p_0(\mathbf{z})}{p_1(\mathbf{x})} = \frac{q_0(\mathbf{z})}{q_1(\mathbf{x})} = \left| \det \frac{\partial \mathbf{x}}{\partial \mathbf{z}} \right| \quad \text{(The Jacobian is independent of the distribution.)}$$

where

$$p_0(\mathbf{z}) = \frac{1}{\sqrt{2\pi\sigma_0^2}} \exp\left(-\frac{(\mathbf{z}-\mu_0)^2}{2\sigma_0^2}\right)$$

$$p_1(\mathbf{x}) = a\frac{1}{\sqrt{2\pi\sigma_1^2}} \exp\left(-\frac{(\mathbf{x}-\mu_1)^2}{2\sigma_1^2}\right) + (1-a)\frac{1}{\sqrt{2\pi\sigma_2^2}} \exp\left(-\frac{(\mathbf{x}-\mu_2)^2}{2\sigma_2^2}\right)$$

$$q_0(\mathbf{z}) = \frac{1}{\sqrt{2\pi\sigma_3^2}} \exp\left(-\frac{(\mathbf{z}-\mu_3)^2}{2\sigma_3^2}\right)$$

$$q_1(\mathbf{x}) = \frac{1}{\sqrt{2\pi\sigma_1^2}} \exp\left(-\frac{(\mathbf{x}-\mu_1)^2}{2\sigma_1^2}\right)$$

Rewriting the equality above, we get

$$\frac{p_1(\mathbf{x})}{q_1(\mathbf{x})} = \frac{p_0(\mathbf{z})}{q_0(\mathbf{z})}$$

$$\text{LHS} = \frac{a\frac{1}{\sqrt{2\pi\sigma_1^2}} \exp\left(-\frac{(\mathbf{x}-\mu_1)^2}{2\sigma_1^2}\right) + (1-a)\frac{1}{\sqrt{2\pi\sigma_2^2}} \exp\left(-\frac{(\mathbf{x}-\mu_2)^2}{2\sigma_2^2}\right)}{\frac{1}{\sqrt{2\pi\sigma_1^2}} \exp\left(-\frac{(\mathbf{x}-\mu_1)^2}{2\sigma_1^2}\right)}$$

$$= a + (1-a)\frac{\sigma_1}{\sigma_2} \exp\left(\left(\frac{1}{2\sigma_1^2} - \frac{1}{2\sigma_2^2}\right)\mathbf{x}^2 + c\mathbf{x} + d\right)$$

$$\text{RHS} = \frac{\frac{1}{\sqrt{2\pi\sigma_0^2}} \exp\left(-\frac{(\mathbf{z}-\mu_0)^2}{2\sigma_0^2}\right)}{\frac{1}{\sqrt{2\pi\sigma_3^2}} \exp\left(-\frac{(\mathbf{z}-\mu_3)^2}{2\sigma_3^2}\right)}$$

$$= \frac{\sigma_3}{\sigma_0} \exp\left(\frac{(\mathbf{z}-\mu_3)^2}{2\sigma_3^2} - \frac{(\mathbf{z}-\mu_0)^2}{2\sigma_0^2}\right)$$

$$= \exp(e\mathbf{z}^2 + f\mathbf{z} + g)$$

for some constants $c$, $d$, $e$, $f$ and $g$ where one of $e$ and $f$ is non-zero. Since $\sigma_1^2 < \sigma_2^2$, we know $\left(\frac{1}{2\sigma_1^2} - \frac{1}{2\sigma_2^2}\right)\mathbf{x}^2 + c\mathbf{x} + d \to -\infty$ and LHS$\to a > 0$ as $\mathbf{x} \to \infty$ or $-\infty$. When $\mathbf{x} \to \infty$ or $-\infty$, we have $\mathbf{z} \to \infty$ or $-\infty$ as well since $f$ is bijective on $\mathbb{R}$ and singular. However, the RHS can only converge to 0 or diverge to $\infty$ when taking the limit of $\mathbf{z}$. We get a contradiction.

**Case 2** $\sigma_1^2 = \sigma_2^2$. We also show that the inverse mapping of $X \sim \mathcal{N}(\mu_1, \sigma_1^2)$ is not a Gaussian by contradiction. Following similar steps in **Case 1**, we get

$$LHS = a + (1-a)\frac{\sigma_1}{\sigma_2} \exp\left(c\mathbf{x} + d\right)$$

$$RHS = \exp(e\mathbf{z}^2 + f\mathbf{z} + g)$$

| Model | ↑ LL | ↓ LL score |
|-------|------|-----------|
| PPF-D | $-2.072$ | 0.331 |
| PPF-P | $\geqslant -1.785$ | 0.044 |

Table 4: Log-likelihood comparison of PPF-D and PPF-P. LL score represents the difference of log-likelihood under the true underlying distribution and log-likelihood under the learned model. Arrow (↑)/(↓) shows higher/lower values are better.

for some constant $c$, $d$, $e$, $f$ and $g$ where $c$ must be non-zero and at least one of $e$ and $f$ is non-zero. Taking the limit of $\mathbf{x}$ such that $c\mathbf{x} + b \to -\infty$, we have LHS$\to a$. By the bijectivity and singularity of $f$, we know $\mathbf{z}$ goes to either $\infty$ or $-\infty$. However, in either case, the RHS can only diverge to $\infty$ or converge to 0. We get a contradiction. □

To complement Proposition 1, we generate sequential data with the following property to train PPF-D and PPF-P models to fit the data: the underlying distribution of values at each time-step switches between a mixture of Gaussians distribution and one component of the mixture distribution. More specifically, the underlying distribution of observations at even time steps follows $\mathcal{N}(4, 1)$ and the underlying distribution for odd time steps follow a Gaussian mixture distribution of two components $\mathcal{N}(4, 1)$ and $\mathcal{N}(10, 1)$ with equal weights. We created a dataset of 1000 sequences where each sequence has the length of 15, and trained both PPF-D and PPF-P on this dataset.

Figure 3 shows the experimental results for this experiment. We can see that PPF-D is not able to handle data generated by this distribution very well. The output distribution of PPF-D tries to cover both components of the Gaussian mixture distribution, but most of the samples are concentrated in an area of low probability. In contrast, we can see that PPF-P, with more flexible base distribution, is much better at modeling sequences sampled from our synthetic switching distribution. Most of the data sampled from PPF-P model lie in the high-probability region: At odd time step, the sampled data can hit both components of the Gaussian mixture model and at even time step, the sampled data can also recover the ground truth distribution. Table 4 shows the log-likelihood of test sequences under the distribution learned from our model vs. the log-likelihood under the true distribution. The better estimation of PPF-P is conformed by a higher log-likelihood. We also reported the difference of log-likelihood under the true distribution and log-likelihood under the learned model. PPF-P has a lower score which shows it performs better in estimating the true underlying distribution.

