# OpenReview forum: "Point Process Flows"
_ICLR.cc/2020/Conference — Reject_

### Official Review · AnonReviewer1 · 2019-10-22
**Official Blind Review #1**

**Rating:** 3

**Review:**

This paper proposes a model for point processes using normalizing flows. The conditional distributions of next inter-arrival times are modeled as normalizing flows with base distributions conditioned on the history. Continuous-time flow based on neural ODE was employed.

Overall I find this paper incremental. There have been several works using deep generative models to temporal data, and the proposed method is a simple combination of well-established existing works without problem-specific adaptation.

I don’t get the point of using VAE in the likelihood estimation. The model (PPF-D) already defines a flexible conditional distribution. The only reason I can imagine to introduce VAE type model is when the dimension of the latent variables (which should be strictly the same as the observed variable) is too large to model directly. In such case one may choose to optimize a lower bound computed from the variational distribution defined over the lower-dimensional latent spaces. Hence in case of temporal point processes where the dimension is one, I see no point of doing VAE. The authors stated that VAE based model (PPF-P) performs better than ordinary flow model (PPF-D) because PPF-P has more flexible base distribution. My guess is that PPF-P has one more stochastic layer so the architecture itself is more expressive than PPF-D. PPF-D with more complex structure (e.g., more layers in base distribution parameter network) may result in a similar performance.

The authors stated in coclusion section that “The proposed PPF can be optimized by maximizing the exact likelihood”, which is not true for PPF-P optimizing the lower bound.

**Experience Assessment:**

I have read many papers in this area.

**Review Assessment: Checking Correctness Of Derivations And Theory:**

N/A

**Review Assessment: Checking Correctness Of Experiments:**

I assessed the sensibility of the experiments.

**Review Assessment: Thoroughness In Paper Reading:**

I read the paper at least twice and used my best judgement in assessing the paper.

---

> ### Author Response · Authors · 2019-11-15
> **Response to Review #1 - part 1**
>
> Thank you for your time and review. We would like to clarify main concerns and raised questions.
>
> C1. Overall I find this paper incremental. There have been several works using deep generative models to temporal data, and the proposed method is a simple combination of well-established existing works without problem-specific adaptation.
>
> R1. We would like to clarify our contribution and the importance of employing each technique in modeling point process distributions.
> In this paper, we proposed a new perspective on modelling point process distributions which directly estimates the density of conditional point process distribution by utilizing normalizing flow. The main advantage of this method in comparison to the intensity-based methods is that we are able to model arbitrary complex point process distributions without making any parametric assumption on the functional form of the distribution while being able to evaluate the likelihood.
> In order to make our model more expressive and flexible, we proposed to have base-distributions with probabilistic parameters by utilizing variational auto-encoder paradigm. When using flow techniques for density estimation, the expressiveness of the model is not only limited by the complexity of normalizing flow transformations, but also by the class of base-distributions. In our proposed framework, the fact that flow transformations are shared across time-steps and the true underlying distribution across time-steps might vary a lot, makes our model more sensitive to the choice of base distribution family. We believe that, If we choose to model base distributions with Gaussian distribution with deterministic parameters, the bijective transformation might not be able to estimate underlying distributions well, especially when the the ground-truth target distributions vary a lot across time-steps of all the sequences.  We further support our claim by proving Proposition 1 in Appendix A which, intuitively speaking, says more flexible base-distribution yields more expressive model.
> In order to address this shortcoming, we proposed more flexible base distributions where the parameters are probabilistically modeled using variational auto-encoder framework. The flexibility comes from the fact that by marginalizing over the latent space of VAE  $p(z_n^{\tau})=\int_{z_n^{vae}}^{ }p_\rho(z_n^{\tau}|z_n^{vae}) dz_n^{vae}$, the base distribution could be highly flexible. The base distribution of $z_n^\tau$ follows different Gaussian distributions conditioned on different samples of $z_n^{vae}$.
>
> To demonstrate the power of our proposed probabilistically modeled base-distribution , we designed a synthetic experiment as follows:
>
> -     We construct a dataset of sequences, where in each sequence, the underlying distribution at each time-step varies from a unimodal Gaussian distribution $p(x)=N(4,1)$ (at odd time steps) to a bimodal Gaussian distribution $p(x) = ½*N(4,1)+½*N(10,1)$ (at even time steps).
> -    We chose one mode (N(4,1)) to be the same between even and odd time steps.
> -     The motivation for this experiment is as follows. For PPF-D, by modelling the base distribution with the deterministic parameters , in our example there is no one-to-one transformation that can map two Gaussian base distributions exactly into a uni-modal and a multi-modal distribution respectively at the same time. In other words, if a singular one-to-one mapping can map a Gaussian distribution to the mixture of Gaussian distribution in our example, the inverse mapping of either component of the mixture of Gaussians can not be a Gaussian. Please see  Preposition 1 and its proof that are provided in Appendix A.
> -     On the contrary, PPF-P, with a more flexible base-distribution, should be able to better estimate the true distribution.
> -     We visualized  samples generated by both PPF-P and PPF-D for an even and odd time-step. The results illustrate when the true underlying distribution is mixture of Gaussians, PPF-D covers both components of the kernels but most of the samples are concentrated in an area of low probability, somewhere in between the means of components of mixture distribution. In contrast, the results show that PPF-P, with a more flexible base-distribution is much better in estimating the true underlying distribution.  Most of the data sampled from PPF-P model lie in the high-probability region.
> -    We reported the log-likelihood of test data for both PPF-P and PPF-D and also reported the difference of log-likelihood under the true distribution and log-likelihood under the learned models (LL score).  The better estimation of PPF-P is confirmed by having a higher log-likelihood and a lower LL score in comparison to PPF-D.
>
> Please see Appendix A for experimental results and more details on this example.

---

> > ### Author Response · Authors · 2019-11-15
> > **Response to Review #1 - part 2**
> >
> >
> > There are a few works on intensity-free modeling of point process distributions. To the best of our knowledge, this is the first work that treats modeling point process distribution as density estimation using normalizing flow technique while being intensity-free and also being able to evaluate likelihood. We agree that normalizing flow and variational auto-encoder are not new but the well-motivated employment of each component is new and results in the better performance of our model over existing ones.
> >
> >
> > C2. The authors stated in conclusion section that “The proposed PPF can be optimized by maximizing the exact likelihood”, which is not true for PPF-P optimizing the lower bound.
> >
> > R2. We would like to thank the reviewer for raising this issue. Using the word ‘exact’ was mis-leading. We’ve revised this in the updated version. For PPF with the probabilistic approach in learning the base distribution parameters (PPF-P), the model is optimized by maximizing a lower-bound on the real likelihood. However, for our PPF with the deterministic approach in learning base distribution parameters (PPF-D), the model is optimized by maximizing the exact likelihood.
> >
> >
> > Please reconsider the rating if we have satisfactorily answered the reviewer's questions.

---

### Official Review · AnonReviewer3 · 2019-10-22
**Official Blind Review #3**

**Rating:** 6

**Review:**

The authors propose a method for learning models for discrete events happening in continuous time by modelling the process as a temporal point process. Instead of learning the conditional intensity for the point process, as is usually the case, the authors instead propose an elegant method based on Normalizing Flows to directly learn the probability distribution of the next time step. To further increase the expressive power of the normalizing flow, they propose using a VAE to learn the underlying input to the "Flow Module". They show by means of extensive experiments on real as well as synthetic data that their approach is able to attain and often surpass state of the art predictive models which rely on parametric modelling of the intensity function. The writers have put their contributions in context well and the presentation of the paper itself is very clear.

Though the final proof is in the pudding, and the addition of the VAE to model the base distribution yields promising results, the only justification for it in the paper is to create a more "expressive" model. There are multiple ways of increasing the expressiveness of the underlying distribution: moving from RNNs to GRU or LSTMs, increasing the hierarchical depth of the recurrence by stacking the layers, increasing the size of the hidden state, more layers before the output layer, etc. A convincing justification behind using a VAE for the task seems to be missing. Also, using the VAE for a predictive task is a little unusual.

Another, relatively small point which the authors glance over is the matter of efficient training. The Neural Hawkes model suffers from slow training because of the inclusion of a sampling step in the likelihood calculation. I believe that since the model proposed by the authors allows easy back-propagation, their model ought to be easy and fast to train as well. Including the training time for the baselines, as well as the method proposed by the authors, will help settle the point.

Minor point:

 - The extension of the method to Marked Temporal Point Processes in the Evaluation section seems out of place, esp. after setting up the expectation that the marks will not be modelled initially, up till footnote 2 on page 7.

**Experience Assessment:**

I have published in this field for several years.

**Review Assessment: Checking Correctness Of Derivations And Theory:**

I assessed the sensibility of the derivations and theory.

**Review Assessment: Checking Correctness Of Experiments:**

I assessed the sensibility of the experiments.

**Review Assessment: Thoroughness In Paper Reading:**

I read the paper at least twice and used my best judgement in assessing the paper.

---

> ### Author Response · Authors · 2019-11-15
> **Response to Review #3 - part 1**
>
> Thank you for your time and review. We would like to clarify main concerns and raised questions.
>
> C1. Motivation behind VAE
>
> R1. Employing the variational auto-encoder framework in our model plays an important role in helping the flow transformations to better estimate the underlying density. When using flow techniques for density estimation, the expressiveness of the model is not only limited by the complexity of normalizing flow transformations, but also by the class of base-distributions. In our proposed framework, the fact that flow transformations are shared across time-steps and the true underlying distribution across time-steps might vary a lot, makes our model more sensitive to the choice of base distribution family. We believe that, if we choose to model base distributions as Gaussian distributions with deterministic parameters, the bijective transformation might not be able to estimate underlying distributions well, especially when the the ground-truth target distributions varies a lot across time-steps of all the sequences.
> In order to address this shortcoming, we proposed more flexible base distributions where the parameters are probabilistically modeled using variational auto-encoder framework. The flexibility comes from the fact that by marginalizing over the latent space of VAE  $p(z_n^{\tau})=\int_{z_n^{vae}}^{ }p_\rho(z_n^{\tau}|z_n^{vae}) dz_n^{vae}$, the base distribution could be highly flexible. The base distribution of $z_n^\tau$ follows different Gaussian distributions conditioned on different samples of $z_n^{vae}$.
>
> To motivate this argument, we designed the following experiment:
>
> Consider a dataset of sequences, where in each sequence, the underlying distribution at each time-step varies from a Gaussian distribution $p(x)=N(4,1)$ to a mixture of Gaussians $p(x) = ½*N(4,1)+½*N(10,1)$ where one mode of mixture of Gaussians has the same mean and variance as the other unimodal Gaussian distribution.  We trained both PPF-D and PPF-P on this synthetic dataset and experimental results show that PPF-D is not able to fully recover the true underlying distribution. The motivation for this experiment is that, in our example by modelling the base distribution with the deterministic parameters, there could be no one-to-one transformation that can map two Gaussian base distributions exactly into a uni-modal and a multi-modal distribution respectively at the same time. In other words, if a singular one-to-one mapping can map a Gaussian distribution to the mixture of Gaussian distribution in our example, the inverse mapping of either component of mixture of Gaussians distribution can not be exactly Gaussian.  This is proved in Proposition 1 in Appendix A.
> We visualized  samples generated by both PPF-P and PPF-D for an even and odd time-step. The results illustrate when the true underlying distribution is a mixture of Gaussians, PPF-D covers both components of the kernels but most of the samples are concentrated in an area of low probability, somewhere in between the means of the two components of mixture distribution. In contrast, the results show that PPF-P, with a more flexible base-distribution, is much better in estimating the true underlying distribution.  Most of the data sampled from PPF-P model lie in the high-probability region.
> We reported the log-likelihood of test data for both PPF-P and PPF-D and also reported the difference of log-likelihood under the true distribution and log-likelihood under the learned models (LL score).  The better estimation of PPF-P is confirmed by having a higher log-likelihood and a lower LL score in comparison to PPF-D. Please see Appendix A for experimental results and more details on this example.
>
> C2. Comparison of Run-time
>
> R2. We thank the reviewer for suggesting the running-time evaluation experiment. Here we report the running time of passing a single sequence of IP process with 21 events to different models for both training and inference time. At inference time of all models, at each time-step,  we sample 1500 inter-arrival times from the predicted distribution. The reported times are average of execution time for 50 trials.
>
>                   Training time (s)    Inference Time (s)
> PPF-P            0.150244                  0.041857
> APP-VAE       0.054544                  0.076158
> APP-LSTM    0.054506                  0.037435
>
> We do not report the running time of PPF-D as the code for PPF-D was not fully optimized to get the minimum training time possible. For the camera-ready version, we will optimize the PPF-D code and the results to the table.

---

> > ### Author Response · Authors · 2019-11-15
> > **Response to Review #3 - part 2**
> >
> >
> > C3. The extension of the method to Marked Temporal Point Processes in the Evaluation section seems out of place, esp. after setting up the expectation that the marks will not be modelled initially, up till footnote 2 on page 7.
> >
> > R3. We would like to thank the reviewer for raising this issue. We revised this in the updated version. For the experiment on breakfast dataset, we used their proposed approach which uses both time and mark data. Our goal is to only compare the performance on modeling inter-arrival times, but to have a fair comparison on this experiment, our model also used the mark information; using mark data has the potential to help capturing the temporal dependencies.

---

### Official Review · AnonReviewer2 · 2019-11-04
**Official Blind Review #2**

**Rating:** 3

**Review:**

The paper proposes a new intensity-free model for temporal point processes based on continuous normalizing flows and VAEs. Intensity-free methods are an interesting alternative to standard approaches for TPPs and fit well into ICLR.

The paper is written well and is mostly good to follow (although it would be good to integrate Appendix A.1 into the main text). The paper proposes interesting ideas to learn non-parametric distributions over event sequences using CNFs and the initial experimental results are indeed promising. However, I found the presentation of the new framework and the associated contributions somewhat insufficient.

The proposed approach seems to consist mostly of applications of existing techniques and of only few technical contributions. There is also no real theoretical analysis of the advantages of the new approach beyond general statements. In addition, the experimental analysis is missing comparisons to
- other intensity-free methods (e.g., [1, 2])
- other NeuralODE based methods (e.g, [3, 4])
and would also benefit from a closer analysis of the models advantages and/or additional tasks. While each of these points on its own would not be very severe, I found that the combination of all of them is problematic in the current version of the paper. I hope that the authors can address this in their response or future revision.

Further comments:
The results on Breakfast of the competing methods seem quite lower than the results published in (Mehrasa 2019). What is the cause for the differences here? For instance, APP-VAE in (Mehsara 2019) would outperform the results of PPF-P both in terms of LL and MAE (142.7 vs 204.9)?

[1] Xiao et al: Wasserstein Learning of deep generative point process models, 2017.
[2] Xiao et al: Learning conditional generative models of temporal points processes, 2018.
[3] Chen et al: Neural Ordinary Differential Equations.
[4] Jia et al: Neural Jump Stochastic Differential Equations

**Experience Assessment:**

I have read many papers in this area.

**Review Assessment: Checking Correctness Of Derivations And Theory:**

N/A

**Review Assessment: Checking Correctness Of Experiments:**

I assessed the sensibility of the experiments.

**Review Assessment: Thoroughness In Paper Reading:**

I read the paper at least twice and used my best judgement in assessing the paper.

---

> ### Author Response · Authors · 2019-11-15
> **Response to Review #2 - part 1**
>
> Thank you for your time and review. We would like to clarify main concerns and raised questions.
>
> C1. The proposed approach seems to consist mostly of applications of existing techniques and of only few technical contributions.
>
> R1. We would like to clarify our contribution and the importance of employing each technique in modeling point process distributions.
> In this paper, we proposed a novel perspective in modelling point process distributions which directly estimates the density of conditional point process distribution by utilizing normalizing flow. The main advantage of this method in comparison to the intensity-based methods is that we are able to model arbitrary complex point process distributions without making any parametric assumption on the functional form of the distribution while being able to evaluate the likelihood.
> In order to make our model more expressive and flexible, we proposed to have base-distributions with probabilistic parameters by utilizing variational auto-encoder paradigm. When using flow techniques for density estimation, the expressiveness of the model is not only limited by the complexity of normalizing flow transformations, but also by the class of base-distributions. In our proposed framework, the fact that flow transformations are shared across time-steps and the true underlying distribution across time-steps might vary a lot, makes our model more sensitive to the choice of base distribution family. We believe that, If we choose to model base distributions as Gaussian distribution with deterministic parameters, the bijective transformation might not be able to estimate underlying distributions well, especially when the the ground-truth target distributions vary a lot across time-steps of all the sequences.  We further support our claim by proving Proposition 1 in Appendix A which, intuitively speaking, says more flexible base-distribution yields more expressive model.
>
> In order to address this shortcoming, we proposed more flexible base distributions where the parameters are probabilistically modeled using variational auto-encoders. The flexibility comes from the fact that by marginalizing over the latent space of VAE  $p(z_n^{\tau})=\int_{z_n^{vae}}^{ }p_\rho(z_n^{\tau}|z_n^{vae}) dz_n^{vae}$, the base distribution could be highly flexible.
> There are a few works on intensity-free modeling of point process distributions. To the best of our knowledge, this is the first work that treats modeling point process distribution as density estimation using normalizing flow technique while being intensity-free and also being able to evaluate likelihood. We agree that normalizing flow and variational auto-encoder are not new but the well-motivated employment of each component is new and results in the better performance of our model over existing ones.
>
> C2. In addition, the experimental analysis is missing comparisons to - other intensity-free methods (e.g., [1, 2]) - other NeuralODE based methods (e.g, [3, 4])
>
> R2. We thank the reviewer for suggesting new baseline models. Here we discuss the advantages of our model in comparison to suggested methods:
>
> Both [1,2] are intensity-free generative adversarial network based models which are optimized by trying to generate sequences of samples that are indistinguishable from the ground-truth sequences. Because of training criterias of GAN models, they result in intractable likelihood.
>
> Both [3, 4] are NeuralODE based models learning a latent representation that helps to better estimate the intensity function. Here we treat learning point process distribution as density estimation which we believe would open up a new perspective on modeling point processes.
>
> Unfortunately, because of the short time limit of rebuttal period, we were not able to finish the experiment comparing the suggested baselines with our approach. We will add the experimental results in camera-ready version.

---

> > ### Author Response · Authors · 2019-11-15
> > **Response to Review #2 - part 2**
> >
> >
> > C3. The results on Breakfast of the competing methods seem quite lower than the results published in (Mehrasa 2019). What is the cause for the differences here?
> >
> > R3. We implemented all the baselines to share the same pipeline. They all share the same data per-processing, training criteria and evaluation metrics to make sure we have a fair comparison. We believe that the difference in the reported log-likelihood is because of different implementation of the evaluation metric. We evaluate the log likelihood averaged over the total number of time-steps instead of the total number of sequences. Our implementation takes the difference of length between sequences into consideration and it is more commonly used in time-series data literature.
> > Furthermore, our point estimation strategy in computing the MAE is slightly different from APP-VAE paper. Given a sequence at time-step $i$, APP-VAE compute the MAE by first sampling from the latent space of VAE $z^{vae}_{i}\sim p_\psi(z^{vae}_{i}|\tau_{1:i-1})$. Then, each of the samples are passed to the decoder and for each sample, they compare the expected value of the predicted distribution with the ground-truth inter-arrival time and report the average of MAEs across samples as the MAE for this time-step as follows:
> > $$
> >     {\mathop{E}}_{z^{vae}_i\sim p_{\psi}(z^{vae}_i|\tau_{1:i-1})}\Big[| {\mathop{{E}}}_{\tau_i \sim p(\tau_i|z_i^{vae})}(\tau_i ) - \tau^{*}_i |\Big]
> > $$
> >
> > where $\tau_i^*$ is the ground-truth inter-arrival at time-step $i$. This approach might not be ideal, when the underlying distribution is multimodal the expected value of the distribution might be from a low-probability region. Instead, our approach uses the absolute error between the sampled time intervals and the ground truth as a proxy of the sampled sequence. Therefore, MAE reflects the mean quality of the sampled sequence.
> >
> > Please reconsider the rating if we have satisfactorily answered the reviewer's questions.

---

### Public Comment · ~Ifigeneia_Apostolopoulou1 · 2019-10-18
**Implementation Details**

Could the authors please provide more implementation details regarding the normalizing flow used in the paper? In contrast to the construction of the base distribution, the other second component of the model (the normalizing flow) is not discussed at all.

Also in the experimental section, it would be very illustrating if the log-likelihood of the real inhomogeneous and self-exciting process was also mentioned (for which there is closed form), in addition to the one obtained by the different models, so that one can evaluate how well the models can recover the true underlying point processes from some of their realizations.

---

> ### Author Response · Authors · 2019-10-25
> **Reply to Implementation Details**
>
> Thank you for your interest and comments. We will update the paper with additional experiments and more details on the implementation details soon.
>
>
>  (1) Implementation details:
>  Please see Appendix A for the architecture details. The flow module $g_\theta$ is a neural ODE model described in subsection 2.3, where the derivative function $h(.)$ is modeled by a multi-layer perceptron (MLP) with $tanh$ activation function. We used the Neural ODE implementation provided by Grathwohl et al. (2019) and Chen et al. 2018. To make sure that our model generates positive inter-arrival times $\tau \in R^+$, we apply an additional exponential transformation $\tau_n= \exp \circ g_{\theta}(z_n)$  and accordingly, the Jacobian term is added to training objective and evaluation criterion. We use one stack of MLP with 3 hidden layers each with 64 dimensions for the neural ODE model $g_\theta$.
>
> (2) Log-likelihood under the true distribution:
> In our experimental setup, we are unable to directly compute the expected log-likelihood of test sequences in a closed form. We created the synthetic datasets with a constraint on the length of sequences. When we simulated sequences, we rejected samples where the length is shorter than 60 given a simulation time limit. Therefore, our sequences were actually generated by a conditional distribution which can not be computed in a closed form. We are running experiments on synthetic datasets generated without the constraint to provide a meaningful comparison between the log-likelihood values of test sequences under our model distribution and also under the true distribution. We will update the paper with the additional experiments soon.

---

### Decision · Program_Chairs · 2019-12-19

**Decision:**

Reject

**Comment:**

The paper proposed to use  normalizing flow to model point processes. However, the reviews find that the paper is incremental. There have been several works using deep generative models to temporal data, and the proposed method is a simple combination of well-established existing works without problem-specific adaptation.